# Anticoagulant Properties of Coated Fe-Pd Ferromagnetic Shape Memory Ribbons

**DOI:** 10.3390/ijms24032452

**Published:** 2023-01-26

**Authors:** Alexander Bunge, Alexandru Chiriac, Mihaela Sofronie, Izabell Crăciunescu, Alin Sebastian Porav, Rodica Turcu

**Affiliations:** 1National Institute R&D for Isotopic and Molecular Technology, 67-103 Donat Street, 400293 Cluj-Napoca, Romania; 2Department of Neurosurgery, “Grigore T. Popa” University of Medicine and Pharmacy, Str. Universității nr. 16, 700115 Iasi, Romania; 3National Institute of Materials Physics, Atomistilor Street 405A, 077125 Magurele, Romania

**Keywords:** anticoagulant, sulfated pectin, ferromagnetic shape memory alloy, melt spinning, iron–palladium alloy

## Abstract

Shape memory alloys, especially ferromagnetic shape memory alloys, are interesting new materials for the manufacturing of stents. Iron–palladium alloys in particular can be used to manufacture self-expanding temporary stents due to their optimum rate of degradation, which is between that of magnesium and pure iron, two metals commonly used in temporary stent research. In order to avoid blood clotting upon the introduction of the stent, they are often coated with anticoagulants. In this study, sulfated pectin, a heparin mimetic, was synthesized in different ways and used as coating on multiple iron–palladium alloys. The static and dynamic prothrombin time (PT) and activated partial thromboplastin time (APTT) of the prepared materials were compared to samples uncoated or coated with polyethylene glycol. While no large differences were observed in the prothrombin time measurements, the activated partial thromboplastin time increased significantly with all alloys coated with sulfated pectin. Aside from that, sulfated pectin synthesized by different methods also caused slight changes in the activated partial thromboplastin time. These findings show that iron–palladium alloys can be coated with anticoagulants to improve their utility as material for temporary stents. Sulfated pectin was characterized by nuclear magnetic resonance (NMR) and Fourier-transform infrared (FTIR) spectroscopy, and the coated alloys by scanning electron microscopy (SEM) and energy dispersive X-ray analysis (EDX).

## 1. Introduction

Ferromagnetic/magnetic shape memory alloys (FSMA/MSMA) are a specific type of shape memory alloys (SMA) that can change their shape when exposed to a magnetic field, in addition to the normal thermal shape memory effect of SMAs. Unlike the thermal shape memory effect, which requires a phase transition of martensite to austenite and back, the magnetic shape memory effect occurs in the martensite phase. The magnetically inducible reproducible strains in FSMA can be orders of magnitude higher than in other magnetostrictive materials as well as (electrically induced) piezoelectric materials and are on a faster timescale than the thermal shape memory effect. In addition to applications for SMA such as mini actuators [1], seismic control [2], aerospace, medical or automobile applications [3], FSMA can have further use for sensors, smart materials and actuators [4]. For example, the FSMA NiMnGa was used to build a micropump for drug delivery in rats [5].

One medical application where SMA can play a large role in the future is stents. These stents are nowadays normally made of steel or polymers and need to be expanded for use. SMA can, for this purpose, act as self-expanding stents, a function that has already been introduced in 2003 using Nitinol, which is a Ni-Ti alloy [6]. Nitinol shows excellent biocompatibility and corrosion resistance. However, it has been found that corrosion resistance in stents can actually be counterproductive, causing, for example, thrombosis, irritation or inflammation [7]. For this reason, materials for temporary stents have been developed, mainly based on magnesium and iron [8]. Unfortunately, while the corrosion rate of magnesium is far too quick, iron degrades a bit too slow when used as a coronary stent. NiMnGa alloys, the most well-known FSMA, also show a too high corrosion rate, in addition to their poor biocompatibility [9]. However Fe-Pd alloys have a decreased half life compared to pure iron due to microgalvanic corrosion, which is still long enough for its intended application [10]. Combined with the excellent biocompatibility and FSMA properties [11,12], these alloys would make good candidates for temporary self-expanding coronary stents.

Melt-spinning is a nonconventional technique where the rapid cooling of the material provides several advantages for the material created, such as the ability to form metastable phases, decreasing the phase segregation and producing relatively thin ribbons. Because of this, it has been considered as a method to produce SMA materials for medical applications [13]. Fe-Pd alloys have often been prepared in this manner due to the above listed advantages [14,15].

A stent introduced into the human body is a foreign object. Either by itself or by the slight involuntary damaging of the arteries upon the introduction of the stent, blood clotting can occur. In order to prevent this, stents are often coated with an anticoagulant substance [16]. This is usually performed in one of two ways: by covalently or ionically binding the anticoagulant to the stent or by capturing the anticoagulant, for example, in a polymer or nanotubes, for a slow-release mechanism. Heparin is one of the most commonly used anticoagulants; however, there have been a number of heparin mimetics based on sulfated carbohydrates [17]. Sulfated pectin is one of these and has been synthesized and verified as an anticoagulant multiple times [18,19,20,21]. Polyethylene glycol (PEG) is a polymer that, although nonionic, displays antithrombogenic activity as a coating [22]. So far, there have been only few reports of a coating of Fe-Pd FSMA [23,24], and none with specific anticoagulant coating.

Aside from binary Fe-Pd FSMA, ternary alloys have also been researched [25,26,27,28,29,30]. Fe-Pd FSM alloy forms upon cooling, besides a face-centered tetragonal (fct) reversible martensitic phase also an irreversible body-centered tetragonal (bct) phase that is undesirable. Adding an element such as Ga or Mn can suppress the formation of such a phase [31,32]. Additionally, by adding elements, the martensitic transition temperature (MT) can be tuned to fit the requirements of the specific application (for cardiovascular stents, the ideal transition temperature would be slightly below body temperature, for example) [26,27,28,29,30].

In the framework of this article, melt-spun ribbons of different binary and ternary alloys Fe-Pd/Fe-Pd-X (X = Mn, Ga) were coated for the first time with different potentially anticoagulant coatings: PEG, pectin and sulfated pectin. The sulfated pectin was synthesized using milder methods than the one commonly used for this purpose. The anticoagulant behavior (measured by PT and APTT in static as well as dynamic conditions) was then determined. These tests increase the knowledge about the obtained materials with respect to their future application as temporary stents.

## 2. Results and Discussion

Four samples of FSMA were prepared (see Appendix A for images of the FSMA before thermal treatment). The composition of the samples was chosen in order to assess the influence of the substrate on coating effectiveness as well as anticoagulant properties. FePd_10_ has a composition which was described first for FePd FSMA and which was used in most previous publications to determine the properties of FePd FSMA, serving as a reference. The melt spinning process used here produced FSMA superior to those obtained by other methods for the envisaged application [15]. While reversible phases are formed due to the process, during cooling, an irreversible bct martensitic phase can still form in pure FePd alloys. For this reason, FePdGa_2-15_ was prepared, which can be fully reversibly cooled/reheated [32]. This reversibility is important when the thermal shape memory effect, which all FSMA also possess, and not the magnetic effect is used. In this case, for medical applications such as temporary stents, an MT below body temperature is required (which FePd has), and it is useful for the MT to be close to body temperature so that the surrounding tissue will not be damaged while heating the stent to effect the MT (if it was higher) and that the stent can be placed correctly before the MT occurs (if it was much lower than body temperature). This can be achieved by doping with Mn and heat treatment. Thus, samples of FePdMn_3-10_ and FePdMn_3-30_ were also prepared [31].

The samples were identified using SEM and EDX measurements (Figure 1 and Appendix A). It is possible to see the austenitic grains, which have a columnar structure as described before [33]. The elemental composition can be determined by EDX in the case of FePd_10_ (Fe_70.6_Pd_29.4_) and FePdGa_2-15_ (Fe_69.3_Pd_29_Ga_1.7_). For the heat-treated samples, FePdMn_3-10_ and FePdMn_3-30_ EDX is not a good way to determine the composition, because during heat treatment, a Mn-rich and a Pd-rich phase forms [31]. The composition of the as-prepared non-heat-treated ribbon is Fe_67_Pd_30_Mn_3_. Increasing the heating time from 10 (FePdMn_3-10_) to 30 min (FePdMn_3-30_) causes an increase in the formation of Mn-rich precipitates, as well as a doubling in size of the single grains, as discussed in more detail in [31,34].

In order to synthesize sulfated pectins **2**, normally chlorosulfonic acid is used as a reagent. This acid is very corrosive and thus difficult to handle. An alternative for the sulfation of hydroxygroups is sulfamic acid. This is a comparatively very mild and stable reagent, which can be used for catalytic sulfation. The most common catalyst for this reaction is urea. Two synthesis procedures were adapted for the synthesis of sulfated pectin **2** (Figure 1): in the first [35], pectin was reacted with urea and sulfamic acid in a deep eutectic melt, while in the second [36,37], pectin was reacted with sulfamic acid and a smaller amount of urea in diglyme as a solvent. The methods had to be adapted because cellulose sulfate was just collected as a precipitate, which is not possible with pectin. It was decided that the best method of purification would be dialysis. Thus, **2a** was obtained by sulfation in melt, and **2b** by reaction in diglyme. Since pectin can be decomposed by NaOH, which was used in the original procedure, NaHCO_3_ was alternatively used for neutralization to obtain **2c** to see if the final product would show any differences. All products were water soluble and did not show any gelling behavior in water like pectin **1** does. In this manner, sulfated pectins were prepared for the first time using sulfamic acid.

The characterization of the obtained sulfated pectins **2** was conducted with FTIR spectroscopy (Figure 2) and ^13^C NMR. Comparing the FTIR spectra of pectin **1** with sulfated pectins **2,** one can see several differences. Aside from the large band at 1215 cm^−1^, which stems from the sulfate group (*v*(S=O)), the bands at 1422 and 1672 cm^−1^ (*v*(C=O) from carboxylate) were also markedly more pronounced in **2**. The reason for this is most likely a hydrolysis of the methyl ester groups under the reaction conditions (the acid group will be then isolated as salt). Additionally, however, between sulfated pectins **2,** there were some differences, though they were not that immediately obvious. The sulfonate band had different intensities between the samples, which indicates a different degree of sulfation, which is potentially a deciding factor in the anticoagulating effects of the polymer. Additionally, the ratios between the bands at 1012 and 1095 cm^−1^ were different, possibly hinting at differences between hydroxygroups (which itself could be because of different degrees of sulfation). The determination of the amount of sulfate groups was then conducted photometrically and amounted to 2.24 mmol/g for **2a**, 1.44 mmol/g for **2b** and 1.46 mmol/g for **2c**, confirming that **2a** contained the most sulfate groups and that the change between NaOH and NaHCO_3_ in **2b** and **2c** only led to small changes in the degree of sulfation.

The three sulfated pectins were also analyzed with ^13^C NMR spectroscopy. Since pectin is a polymer comprised of several different monomers (galacturonate and methyl galacturonate predominantly, but potentially also other monosaccharides such as galactose, glucose, arabinose and rhamnose) that can all be linked in multiple ways and show different degrees of sulfation, the assignment of the signals in the spectra is difficult. In this case, the spectra were compared to already-published spectra of sulfated citrus pectin [20]. Figure 3 shows the ^13^C spectrum of **2a**, and the other spectra can be found in Appendix A. The variation in the peak shift compared to the literature was likely due to salt formation. Even so, from comparison with the literature, it can be seen that methyl galacturonate was still present (CH_3_-O at 53.4 and 52.9 ppm), and that besides methyl galacturonate, galacturonate also existed (ester and carboxylate peaks at 172.4 and 171.9 ppm) and that sulfation also happened (there were multiple C1 peaks at 99.7–97.7 ppm, and there were also multiple methyl ester groups). The spectra of **2b** and **2c** looked similar. It can, however, also be seen that in **2b** and **2c,** a little solvent (diglyme) remained, despite prolonged drying in vacuum at 70 °C. Additionally, the peaks were shifted a bit compared to **2a**, especially the carboxyl peaks, which can be explained by different salt formation from the work up conditions (the presence of sodium from sodium salts can be seen in the EDX in Appendix A).

The coating of the FSM alloys was performed by simple dip coating of the FSMA ribbons into an aqueous solution of PEG, **1, 2a**, **2b** and **2c**, followed by washing with water and drying. A characterization of the samples was conducted with SEM as well as EDX. From SEM (representative images in Figure 4 and Appendix A), no distinct differences that could be attributed to the coating process were visible. This could have either been due to a failure of the coating or a thin uniform layer. That the coating definitely existed becomes evident upon interpreting the EDX results (Appendix A). While the untreated FSMA ribbons showed either no carbon or carbon within the limit of detection, in the coated samples, the carbon content increased from 2 to 15 wt%. With sulfated pectin, sulfur was also clearly distinguishable, although in a small percentage. Thus, the coating of the FSMA sampled proved successful.

For the anticoagulant activity, the PT and APTT values in both static and dynamic conditions were measured for the coated and uncoated samples (Figure 5 and Appendix A). It can be seen that while there were some differences in the FSMA themselves, on their own there was no relevant anticoagulant effect (Appendix A, entries 3–6). The uncoated sample FePdGa_2-15_ showed a slightly larger increase in APTT and even reached a value slightly higher than normal under dynamic conditions. PEG as a coating was also not increasing the anticoagulant behavior (Appendix A, entries 7–10), and in some cases (entry 8 and 9) it even decreased static APTT compared to the uncoated samples. The fact that there were no clear trends visible between the static and dynamic APTT of the respective samples within one group (uncoated or coated with PEG or pectin **1**) indicates that the measured differences were probably not relevant. However, all three sulfated pectins **2** increased the APTT over the normal range, both in static as well as dynamic conditions. The difference between the FSMA substrates was not that large, and while there were no large differences between the different sulfated pectins, on average the anticoagulant effect increased according to **2a** < **2b** < **2c**. The sulfate groups were essential for the coagulation behavior, as pectin alone as a coating did not show heightened APTT (Appendix A, entries 11–14). In neither case did the PT values significantly increase over a normal range. APTT is a measure of coagulation via the intrinsic pathway, whereas PT measures the coagulation via the extrinsic pathway. It is likely that the sulfated pectins influenced mainly the intrinsic pathway and thus the PT remained more or less the same. This was also shown in other publications [20], where the amount of sulfated pectin needed to significantly increase PT was almost an order of magnitude larger than for APTT. Since the intrinsic pathway is considered the more critical one for thrombosis associated with implants [38], the negligible effect of the coatings on PT is not that important.

## 3. Materials and Methods

### 3.1. Materials

Citrus pectin was purchased from Lechner és Zentai Kft., Budapest, Hungary; sulfamic acid was purchased from Silal Trading Srl., Bucharest, Romania; polyethylene glycol 2000 and Fe, Pd, Mn and Ga high-purity elements (99.99%) were purchased from Alfa Aesar, Karlsruhe, Germany; and all other chemicals were purchased from Sigma-Aldrich, Steinheim, Germany.

### 3.2. Methods

The ingots of Fe_70_Pd_30,_ Fe_67_Pd_30_Mn_3_ and Fe_67.5_Pd_30.5_Ga_2_ alloys were prepared by arc melting (under an Ar protective atmosphere) of high purity elements, and the procedure was repeated several times to ensure homogeneity. By using the melt-spinning technique, the ingots were melt spun into ribbons (10–20 mm × 3 mm × 25–30 µm). The speed of the copper wheel was 20 m/s, with a 0.5 mm nozzle diameter and 50 kPa Ar overpressure. The as-prepared ribbons were then treated at 950 °C in vacuum quartz ampoules for different times and were subsequently quenched in ice water. The samples were named FePd_x_ (x = 10 min), FePdMn_3-x_ (x = 15 and 30 min) and FePdGa_2-x_ (x = 15 min), with x—length of thermal treatment at 950 °C.

The synthesis of sulfated pectin **2a**: Citrus pectin **1** (1 g), urea (6 g, 99.9 mmol) and sulfamic acid (5 g, 51.5 mmol) were stirred at 80 °C until a clear greenish solution formed. The reaction was then further stirred in a 150 °C oil bath for 30 min, at which time the solution turned brown and bubbled. After cooling and dissolving in the minimum amount of water (ca. 30 mL), the impure product was dialyzed against water (3 × 6 L, MWCO 2000 Da) and freeze dried to obtain 1.13 g of a brown solid.

Synthesis of sulfated pectin **2b**: Citrus pectin **1** (1.62 g), urea (1.8 g, 30 mmol) and sulfamic acid (3.4 g, 35 mmol) were stirred with diglyme (40 mL) at 160 °C (oil bath temperature) for 15 min. The mixture changed colors from milky white to green to black (bubbling) and dark brown. After refluxing for 1 h, the solvent was evaporated and the remains were redissolved in water, and the solution was basified (pH 12) with NaOH, dialyzed against water and lyophilized to give 1.484 g of a brown solid. For **2c**, the solution was basified with NaHCO_3_ to pH 8 instead of NaOH, and 1.994 g of a brownish solid were obtained.

Coating of FSMA ribbons: An aqueous solution of a 0.5 g coating substance (PEG, **1**, **2a**, **2b** or **2c**) in 25 mL of water was prepared and the ribbons were dipped into it and left without stirring for 17 h. The coated ribbons were then washed with water and ethanol and dried at 60 °C. The respective coated samples were named with the label for the substrate followed by PEG, **1**, **2a**, **2b** or **2c**, respectively.

The determination of sulfate was performed photometrically according to a modified procedure from Silvestri et al. [39], with potassium rhodizonate synthesized according to [40]. ^13^C NMR spectra were recorded on a 500 MHz spectrometer (Bruker Avance III spectrometer) at 125 MHz at room temperature with 45,065 scans in D_2_O.

Anticoagulant assays: Platelet-poor plasma was obtained by centrifuging (3000 g, 10 min, 20–22 °C) fresh human blood (obtained by the blood donation research program of the Romanian National Blood Institute). The plasma (1 mL) with 0.106 M citrate was incubated at 37 °C with the test samples for 30 min, either without (static) or with (dynamic) shaking, cooled and kept on ice. The activated partial thromboplastin time (aPTT) and prothrombin time (PT) for both static and dynamic conditions were measured using an Abrazo Cascade System Helena photo-optic anticoagulation analyzer, according to the instructions of the supplier. All experiments involving human blood were conducted in accordance with the laws and guidelines of Grigore T. Popa University and were approved by the Ethics Committee of this university.

Samples provided for SEM analysis were prepared in the same fashion. Briefly, the samples were deposited on carbon adhesive disks attached to aluminum stubs and were analyzed uncoated at 30 kV in high mag. mode. Image acquisition was conducted on a Hitachi SU8230 High Resolution Scanning Electron Microscope equipped with a cold field emission gun and an 80 X-Max system from Oxford Ins. for EDX analysis. Approximately 90% of the sample was scanned to give a realistic overview of its morphology, and only a few representative areas were captured. Fourier transform infrared (FT-IR) spectra were recorded using a JASCO FTIR 4600A spectrophotometer with an ATR-PRO-ONE accessory. The spectra were CO_2_-, H_2_O-, ATR- and baseline-corrected as well as normalized for better visibility of the bands.

## 4. Conclusions

Sulfated pectin materials were synthesized via new methods using the milder reagent sulfamic acid. Sulfation occurred in all three different syntheses, with sulfate contents of 1.44–2.44 mmol/g as proven by FTIR and ^13^C-NMR. Several Fe/Pd-based ferromagnetic shape memory alloys prepared via melt spinning, which showed promise to be used as temporary stents, were then coated with potential anticoagulants for the first time. The successful coating was proven by EDX. Their anticoagulant activity was then investigated, and it was found that all sulfated pectins increased APTT both in static as well as in dynamic experiments above normal values. These results further demonstrate the usefulness of FSMA for biomedical applications, especially for temporary cardiovascular stents. They can be used in the development of devices that can then be tested in this regard in the future.

## Data Availability

Not applicable.

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
