# Peer review of "Anticoagulant Properties of Coated Fe-Pd Ferromagnetic Shape Memory Ribbons"

_ijms, 2023, doi:10.3390/ijms24032452_

Round 1

Reviewer 1 Report

Abbreviations should be avoided in Abstract.

Nitinol means Nickel Titenium Naval Ordnance Laboratory not NiTi Alloy.

The objective of this article in the introduction has not been properly justified in terms of the state of the art.

Should the commas in Figure 2 and Table 1 be converted into dots?

The description of the stents for the human body is briefly presented in the introduction. Therefore, the reader may expect the developed SMA to be used in the human body. However, this is not the case. Therefore, either the introduction should be changed or the implementation of SMA in human body should be highlighted.

Reviewer 2 Report

This article focuses on the coagulation of Fe-Pd based alloys with different coating. It follows a smooth big logic, but the evidence is not quite enough. Basically, it is better to explain opinions with some evidence, either theories or experiments. The microstructures of different compositions are very different but they are ignored. As the composition doesn’t remain uniform with different phases, will this introduce change to the experiment. For example, FePdGa2-15 is chosen to give fully reversible transition in this article. What is the result? Is it tested and how to do it?

1.     Please state why MT should be lower than body temperature and be close to it.

2.     With different heat treatments time, FePdMn3-10 and FePdMn3-30 show clear difference, please state the reason and microstructure of them.

3.     In 13C-NMR part, the author mentioned formation of different salt, please prove it.

4.     The coated and uncoated SEM shows large difference, not similar. Were these images gotten by Quasi-in situ method or just different samples?

5.     The author uses EDS to show the finish of coating by comparing light element C, is it proper to use EDS to test Carbon?

6.     Please show the overall view of fabricated materials by melt-spinning.

7.     Considering Fe-Pd is shape memory alloys, how the coagulation changes after shape change (strain)?

8.     Data in table 1 could be plotted to give a better comparation.

Reviewer 3 Report

The authors have tried to synthesize some kinds of sulfated pectin in different ways and use them as coating on multiple Fe-Pd alloys in this study. They found that Fe-Pd alloys can be coated with anticoagulants to improve their utility as material for temporary stents via some experiments. The article is innovative and demonstrate the usefulness of FSMA for biomedical applications. However, I think article in this form cannot be published in IJMS yet, the main reasons are:

1. The writing of this article is lack of logical. The part of “Materials and Methods” should be mentioned before you give the readers some results. Besides, there are many sentences irrelevant to the article that need to be selectively retained. You should pay more attention to the writing of the article.

2. You mentioned pectin 2 in Line 113 of the manuscript, but I cant find pectin 2 in the previous part of the article. You should explain Scheme 1 before pectin 2 first appears.

3. Where is Fig. 4 (Line 170)?

4. You should explain more for Table 1 by comparing of each sample, because this part should be the most important part of this article.

5. Try to add a Flow Chart by some pictures for Materials and Methods. This will make the article more readable

6. Because of your writing style, the innovation of the article is not very prominent. Please pay more attention.

7. Can you add some animal experiments?

Round 2

Reviewer 2 Report

The manuscript has been well improved and can be accept now.